# Microbial-Derived Toll-like Receptor Agonism in Cancer Treatment and Progression

**DOI:** 10.3390/cancers14122923

**Published:** 2022-06-14

**Authors:** Eileena F. Giurini, Mary Beth Madonna, Andrew Zloza, Kajal H. Gupta

**Affiliations:** 1Division of Hematology, Oncology, and Cell Therapy, Department of Internal Medicine, Rush University Medical Center, Chicago, IL 60612, USA; eileena_f_giurini@rush.edu (E.F.G.); andrew_zloza@rush.edu (A.Z.); 2Division of Translational and Precision Medicine, Department of Internal Medicine, Rush University Medical Center, Chicago, IL 60612, USA; 3Division of Pediatric Surgery, Department of Surgery, Rush University Medical Center, Chicago, IL 60612, USA; marybeth_madonna@rush.edu; 4Division of Surgical Oncology, Department of Surgery, Rush University Medical Center, Chicago, IL 60612, USA

**Keywords:** innate receptors, toll like receptors, cancer, immunotherapy, microbial based therapy

## Abstract

**Simple Summary:**

Toll like receptors (TLRs) are a group of transmembrane receptors belonging to the class of pattern recognition receptors (PRR), which are involved in recognition of pathogen associated molecular patterns (PAMPs), inducing immune response. During the past decade, a number of preclinical and clinical breakthroughs in the field of TLR agonists has immerged in cancer research and some of these agents have performed exceptionally well in clinical trials. Based on evidence from scientific studies, we draw attention to several microbial based TLR agonists and discuss their relevance in various cancer and explore various microbial based TLR agonists for developing effective immunotherapeutic strategies against cancer.

**Abstract:**

Toll-like receptors (TLRs) are typical transmembrane proteins, which are essential pattern recognition receptors in mediating the effects of innate immunity. TLRs recognize structurally conserved molecules derived from microbes and damage-associated molecular pattern molecules that play an important role in inflammation. Since the first discovery of the Toll receptor by the team of J. Hoffmann in 1996, in *Drosophila melanogaster*, numerous TLRs have been identified across a wide range of invertebrate and vertebrate species. TLR stimulation leads to NF-κB activation and the subsequent production of pro-inflammatory cytokines and chemokines, growth factors and anti-apoptotic proteins. The expression of TLRs has also been observed in many tumors, and their stimulation results in tumor progression or regression, depending on the TLR and tumor type. The anti-tumoral effects can result from the activation of anti-tumoral immune responses and/or the direct induction of tumor cell death. The pro-tumoral effects may be due to inducing tumor cell survival and proliferation or by acting on suppressive or inflammatory immune cells in the tumor microenvironment. The aim of this review is to draw attention to the effects of TLR stimulation in cancer, the activation of various TLRs by microbes in different types of tumors, and, finally, the role of TLRs in anti-cancer immunity and tumor rejection.

## 1. Introduction

The Toll receptor was identified in *Drosophila melanogaster*, a receptor essential for establishing the dorsal–ventral axis during embryonic development [1,2]. Following the discovery of the Toll receptor, several mammalian proteins were found to share structural similarities to the *D. melanogaster* Toll receptor, and thus they were named Toll-like receptors (TLRs) [3]. In 2011, the Nobel Prize in Physiology/Medicine was awarded to Dr. Jules A. Hoffmann and Dr. Bruce A. Beutler who made significant contributions to the discoveries concerning TLRs and their role in innate immunity [4,5]. It was by this discovery that scientists found innate immunity is essential and critical in immune responses to pathogen infections in connection with the activation of adaptive immunity [6].

TLRs comprise a family of type I transmembrane receptors, which are characterized by an extracellular leucine-rich repeat (LRR) domain and an intracellular Toll/IL-1 receptor (TIR) domain [7,8,9]. TLRs trigger immune responses against various invading pathogens by recognizing specific pathogen-associated molecular patterns (PAMPs), which are highly conserved and derived from potential pathogenic microorganisms, such as bacteria, viruses, fungi, and parasites [10,11].

Microbial sensing through TLRs initiates a signaling cascade that induces pro-inflammatory responses [10]. TLRs recruit TIR-specific domain-containing adaptor proteins for the activation of downstream signaling. The main domains recruited by TLRs are the myeloid differentiation factor-88 (MyD88), the Toll/IL-1 receptor domain adaptor protein, and the TIR-domain-containing adapter-inducing interferon-β (TRIF) [12,13]. Upon activation, various signaling pathways are initiated, which results in a variety of inflammatory cytokines, thus undergoing transcription by the phosphorylation of IkBa to activate NF-κB [14,15]. The multiple signaling pathways contribute to the rapid response of the innate immune system to pathogens. TLRs also regulate adaptive immunity by the activation and maturation of dendritic cells and the production of pro-inflammatory cytokines and chemokines, which induce the proliferation and differentiation of Th1 and Th2 cells [16].

TLR signaling has been well-studied in various diseases, including cancer. TLRs are expressed not only on the surface of immune cells but also on tumor cells [17,18]. TLRs are documented to have pro and anti-tumor responses, though to date, the role of TLR signaling is still not completely understood in cancer progression. This review will describe the usage of bacterial- and viral-derived TLR activation in cancer immunotherapy, TLR expression profile on tumors, and the involvement of TLR signaling in tumor outcomes. We will also discuss the status of research in utilizing TLR agonists as potential therapeutic agents in cancer treatment.

## 2. TLR Localization and Recognition of Microbial Ligands

TLRs are involved in the recognition of microbial exogenously and endogenously derived molecular patterns. This occurs at the plasma membrane and at intracellular compartments, respectively, and thus TLR ligands can be either exogenous or host-derived. TLRs 1, 2, 4, 5, and 6 are located primarily in the plasma membrane, where they interact with components of microbial pathogens that exogenously encounter the cell. In contrast, TLRs 3, 7, 8, and 9 are situated in the membranes of endosomes and lysosomes, where they interact with components that are endogenous. Figure 1 depicts in detail the location of specific TLRs and their respective, best-characterized ligands.

## 3. Expression of TLRs on Tumor Cells and Its Clinical Relevance

TLRs are known to be expressed and activated in innate immune cells, such as macrophages and dendritic cells (DCs); however, in recent years, several studies have shown that TLRs are also highly expressed by various tumor cells. Therefore, the study of TLR expression and function in cancer has become a focus for researchers in field of cancer immunotherapy. Table 1 summarizes the TLRs expressed on various tumor cells in different cancer types along with clinical outcomes.

## 4. Microbial Derived TLR Agonists and Their Role in Cancer Immunotherapy

Harnessing the host response to infection has been utilized to target cancer for centuries [81,82]. Microbial sensing through TLRs initiates a signaling cascade that induces pro-inflammatory responses. The Toll-like receptor signaling pathway plays a crucial role in host immune defenses against numerous diseases and has been identified as an immunotherapeutic target against various types of cancer. New avenues to combat cancers involves the regulation of the host’s innate immune system using agonists, which can bind to a variety of TLRs. These agonists can be used in combination with other cancer therapies, like chemotherapy and radiotherapy, to provide a broader spectrum of protection.

## 5. Bacterial-Derived TLR Agonists

Many of the pivotal studies in the field of bacterial-based cancer immunotherapy (BBCT) were pioneered by 19th century clinician–scientist Dr. William Coley [81,82]. Dr. Coley utilized a combination of live *Streptococcus pyogenes* and *Serratia marcescens* to treat patients with inoperable sarcomas [81]. In the patients with the most dramatic tumor regression, Dr. Coley noted that an erysipelas infection and subsequent fever had been induced in these patients. Dr. Coley’s observations were likely a result of robust immune system activation from Coley’s bacterial-based toxins, thus implicating a link between response to infection and tumor eradication that would serve as the basis for modern cancer immunotherapies. Though unbeknownst to Dr. Coley, a likely mediator in his patients’ responses was innate microbial recognition through Toll-like receptors [82,83].

Molecular triggers responsible for remissions from Coley’s toxins are TLR ligands that target patter recognition receptors (PRRs) [84]. PRRs share the ability to recognize relatively conserved microbial components, which are generally referred to as microbe- or pathogen-associated molecular patterns (MAMPs or PAMPs), as well as endogenous danger signals commonly known as damage-associated molecular patterns (DAMPs). Common TLR-activating MAMPs include viral and bacterial nucleic acids (which can signal through TLR3, TLR7, TLR8, or TLR9), flagellin (a TLR5 agonist), as well as lipopolysaccharide (LPS), lipoteichoic acid, and mannans (which signal through TLR2 or TLR4). Endogenous nucleic acids and the nuclear non-histone protein high mobility group box 1 (HGMB1) are prototypic TLR-activating DAMPs.

Due to their role in self/nonself-differentiation [85] and their ability to induce antigen presenting cell (APC) maturation, TLR agonists are considered promising adjuvant candidates [86]. In fact, a number of TLR agonists, including Pam3CSK4, Pam2CSK4, MPLA (a LPS derivative), CpG, PolyI:C, and flagellin, are currently being tested as cancer agonists [87].

Roberts et al. recently showed that an attenuated strain of *Clostridium novyi* efficiently decreased tumor size in rat and dog cancer models in addition to one sarcoma patient [88]. This treatment is well-targeted as spores of *Clostridium novyi* germinate selectively within the hypoxic regions of cancerous tissue and induce immune responses likely via TLR activation [89].

## 6. Viral-Derived TLR Agonists

Oncolytic viruses (OVs) preferentially target tumor cells and activate antitumor immunity while limiting pathogenicity, thus they have emerged as a promising tool for viral-based cancer therapies [90]. Though additional studies must be conducted to further characterize the TLR contribution to antitumor responses from oncolytic virotherapy, OVs possess several TLR-stimulating moieties that can contribute to activation of host immune response at the tumor site. Intratumoral treatment of murine glioma tumors with oncolytic adenovirus Delta-24-RGD significantly reduced tumor growth compared to PBS-treated tumors [91]. Delta-24-RGD treatment was demonstrated to remodel the tumor microenvironment, predominantly through enhanced CD8+ and CD4+ T cell infiltration in the tumor [91]. Immune-mediated targeting of the tumor is likely a result of TLR9-medated T cell proliferation as well as maturation of APCs, following the TLR9 recognition of double-stranded adenoviral DNA [92,93,94].

Enhancing the intrinsic TLR stimulating properties of oncolytic adenoviruses has been another strategy for targeting TLR-mediated immune responses. Ad5D24-CpG, an oncolytic adenovirus genetically manipulated to express TLR9 stimulating CpG islands, significantly controlled tumor growth compared to CpG unenhanced Ad5D24 treated tumors [95]. Antitumor response from Ad5D24-CpG treatment was determined to be highly reliant on TLR9-mediated NK cell activation, leading to the effective killing of tumor cells [95]. Because stimulation of TLRs expressed on NK cells enhances release of cytotoxic granules and cytokine production, in this context it is suggested that the CpG insertion in Ad5D24 enhances these NK cell properties, facilitating more robust antitumor responses [96,97].

TLRs have also been implicated in the usage of non-oncolytic viruses for viral-based therapeutics. The intratumoral administration of heat-inactivated influenza but not active influenza virus was found to drastically inhibit tumor growth in a B16 melanoma model [98]. This response from heat-inactivated influenza treatment was determined to be dependent on increased cross-presenting CD8+ dendritic cell infiltration in the tumor [98]. Paralleling the in vivo findings, heat-inactivated influenza was observed to more potently stimulate TLR7 compared to active influenza virus, determined by using TLR7 reporter cells [98]. Given that the recognition of TLR ligands can enhance antigen cross presentation in DCs, it is likely that heat-inactivated influenza more potently engaged TLR7, leading to an increase in cross-presenting DCs within the tumor and the activation of CD8+ T cells and thereby reduced tumor progression [99,100,101].

## 7. TLR–TLR Cross-Talk and the Modulation of Immune Response

Targeting the ability of multiple TLRs to synergize with each other has been another strategy in TLR-focused cancer therapies, as it emulates the multi-TLR activation encompassed by an invading pathogen [102,103,104,105]. Mimicking the TLR recognition profile of influenza virus, the intratumoral administration of a TLR3 agonist and a TLR7 agonist has been shown to dramatically reduce tumor progression through increased granzyme B and perforin expression in CD8+ T cells, as well as an increased M1 to M2 macrophage ratio within murine lung tumors [106]. TLR synergy in a cancer context has been demonstrated in studies where the administration of either TLR agonist individually did not elicit the same robust antitumor response as did the combination [107]. Targeting the same microbial-like TLR activation profile has also been demonstrated to be effective in other tumor models. Nanoparticle complexes comprising TLR3 and TLR7 agonists poly(I:C) and imiquimod, respectively, were demonstrated to achieve complete tumor rejection in a B16 melanoma model. Immune memory from treatment with the TLR agonist complex was further demonstrated through a lack of tumor development following rechallenge with B16 melanoma [108].

Viral-like TLR activation for cancer therapeutic applications extends beyond an influenza-mimicking TLR activation profile. DNA viruses, such as herpes simplex virus and vaccinia virus, are also multi-TLR activators, through TLR2 [109] recognition of viral envelope residing in glycoproteins as well as TLR9 [110] activation from unmethylated CpG motifs found in the viral genome [111,112,113,114]. To utilize the synergistic interactions between TLR2 and TLR9, the cervical cancer tumor antigen E7 was combined with TLR2/TLR9 agonists to reverse the immunosuppressive tumor microenvironment in TC-1 tumors. By diminishing myeloid-derived suppressor cell (MDSC) and regulatory T cell populations, the tumor became permissive to antitumor CD8+ T cell infiltration and the immune-mediated suppression of tumor growth [114,115]. TLR2/TLR9 activation has also been shown to enhance DC maturation, which is essential for the effective processing and presentation of tumor antigens on MHC molecules [116,117,118,119]. Targeting DCs with lipoprotein and CpG ODN TLR agonists combined with tumor antigen decreased the production of immunosuppressive cytokine IL-10 while increasing IL-12 production from DCs. The shift in cytokine production has a dual effect on the tumor microenvironment with the decrease in IL-10 production relieving IL-10-mediated suppression of DC function, and the increase in IL-12 production driving cytotoxic T lymphocyte-activating Th1 responses [118,120].

Bacteria also possess multiple structural motifs recognized by TLRs that amplify host inflammatory responses when activated simultaneously. Therapeutic synergy of bacterial ligand sensing TLRs has been demonstrated in D2F2 tumors, following treatment with both CpG ODN and TLR5 agonist flagellin [121]. Tumors treated with CpG ODN or flagellin as a single therapy did not exhibit inhibition of tumor growth, suggesting that signaling from multiple TLRs is needed to initiate robust antitumor responses. Other combinations of TLR agonists mimicking bacterial recognition have been effective in the formation of cancer vaccines when combined with tumor antigens. Mirroring a TLR activation profile of Gram-negative bacteria [122,123], monophosphoryl6 lipid A and CpG ODN were bound to silicified murine ovarian tumor cells as a source of tumor antigen. Mice vaccinated with this combination of tumor cells and TLR agonists experienced significant tumor regression of established high-grade ovarian tumors [124]. Akin to the enhanced DC maturation following bacterial recognition by TLRs, tumor vaccine-treated DCs also experienced increased tumor antigen uptake and MHC expression, contributing to the induction of tumor-specific T cell immunity [124,125,126].

The usage of the TLR4/TLR9 activation profile in cancer vaccines has been extended to clinical trials [127,128,129,130]. AS15, an immunostimulant containing TLR4 and TLR9 agonists combined with tumor antigen MAGE-A3 and high dose IL-2, was assessed for anticancer responses in metastatic melanoma patients. The combination therapy had a disease control rate of 63%, in which 19% of patients experienced a complete response to the combination of therapy. Peripheral blood regulatory T cells from treatment responders had decreased expression in immune checkpoint proteins, such as CTLA-4 and 4-1BB, while expression was increased in non-responders [130].

## 8. TLR Signaling in Cancer

### 8.1. Effects of Tumor-Promoting TLR Signaling

TLRs have been associated with tumorigenesis, as they can activate multiple cancer-associated signaling pathways. To date, TLRs have been recognized to transduce signals through NF-κB, PI3k-Akt, and MAPK-ERK to advance cancer. NF-κB is recognized as the canonical signaling target upon TLR activation. When activated, NF-κB plays a role in several cellular functions, including cell proliferation, pro-inflammatory cytokine production, and cell survival/apoptosis, as described in Figure 2 [131]. As a result of regulating many cellular processes, the NF-κB pathway is an optimal target for aberrant, pro-tumorigenic signaling [132]. Through the LPS stimulation of TLR4, NF-κB activation enhanced the proliferation of gastric cancer cell lines BGC-823 and SGC-7901 [131,133]. In similar form, NF-κB activation via TLR4 resulted in apoptosis evasion in lung and head and neck cancer, with the addition of TNF-related apoptosis-inducing ligand (TRAIL) [134,135].

Tumorigenic TLR signaling extends beyond the canonical NF-κB pathway. Adaptor protein BCAP has been recognized as a link to TLR activation and downstream PI3k-Akt signaling [136,137]. PI3K-Akt signaling regulates cell proliferation, cell growth, and survival, thus is frequently upregulated in many cancers [138] (Figure 3). TLR7 activation was found to perpetuate pancreatic cancer progression through upregulated PI3K/Akt signaling. Consequently, an upregulation of downstream signaling targets was also observed, including antiapoptotic and pro-proliferative genes Bcl-xL and c-Myc [64,134]. Apoptosis resistance in PCI-30 cells has been shown to be mediated by TLR4-PI3K/Akt signaling [135]. Enhanced angiogenesis has also been described as a product of PI3K-Akt signaling, particularly in a cancer context. TLR4 stimulation in PANC-1 cells has increased vasculature formation and proliferation through PI3K/Akt dependent signaling [138].

Aberrant MAPK/ERK signaling has been heavily implicated in approximately a third of all human cancers [139]. Through activation of TAK1, TLR stimulation can activate the MAPK/ERK pathway to regulate growth, cell survival, and metastasis [140] (Figure 4). LPS-induced TLR4 stimulation in A549 and H1299 cells was found to promote secretion of pro-angiogenic factors VEGF and IL-8 in a p38 MAPK dependent manner [141]. MAPK/ERK signaling utilizes transcription regulation to promote tumoral immune evasion [140]. Endogenous activation of TLR2 on glioma-associated microglia was observed to downregulate MHC class II expression and impede antigen presentation in a TLR2-ERK1/2-dependent manner [142,143]. As a result, proliferation and activation of glioma targeting CD4+ T cells was hindered.

### 8.2. Effects of Anti-Tumor TLR Signaling

The fate of pro- or anti-tumor TLR signaling seems to be largely context-dependent, with considerations being the cancer type, the ligand activating the TLR, and the TLR itself. Tumor-rejecting TLR signaling utilizes several of the same pathways that perpetuate tumor growth, including NF-κB. TLR-mediated NF-κB signaling has been shown to induce IL-1β, IL-6, and TNF-α production in breast and bladder cancer models [26,144]. Targeting MAPK/ERK signaling has also been reported to promote anti-tumor responses. Dendritic cells stimulated with TLR3 and TLR7 agonists were found to upregulate ERK signaling, likely contributing to the enhanced dendritic cell activation and anti-tumor T-cell responses observed in CT26 tumors [145].

Apoptosis-induction in cancer cells appears to be a primary target for anti-tumor TLR signaling. Unsurprisingly, modulating NF-κB signaling has been of interest to promote cancer cell apoptosis [131]. The poly(I:C) stimulation of TLR3 on PCI-15B cells was reported to enhance apoptosis through sustained NF-κB inactivation [146]. As a result, inactive NF-κB is likely unable to activate downstream anti-apoptotic genes, such as BCL-2 [131]. In contrast, TLR3-mediated NF-κB activation was found to be required for apoptosis in poly(I:C)-treated CAMA-1 cells [22]. It appears that the activation or inactivation of NF-κB to promote apoptosis is dependent on the cancer type and additional factors that may be driving the cancer.

Beyond NF-κB, the stimulation of TLRs can activate a type I interferon (IFN) signaling pathway to promote anti-tumor responses (Figure 5). Because IFN signaling can occur to induce apoptosis in infected cells, it has been recognized as a beneficial tool for initiating apoptosis in malignant cells. Treating LNCaP cells with a TLR3 agonist was found to promote apoptosis in an IRF-3 signaling-dependent manner. Through TLR–IFN signaling, pro-apoptotic protein Noxa was subsequently upregulated downstream, a likely contributor to the increased cancer cell death [147]. Apoptosis mediated by IFN signaling extends to other cancer types. In a non-muscle invasive bladder cancer model, TLR4 activation with P-MAPA was shown to enhance IFN signaling. TLR4-mediated IFN signaling resulted in increased apoptosis and iNOS expression, an enzyme responsible for nitric oxide (NO) synthesis [144]. Increased intracellular NO concentrations have been shown to induce apoptosis [148], thus it is likely that the apoptosis of bladder cancer tumors occurs through a TLR4–IFN–iNOS axis [149].

## 9. Conclusions and Future Directions

Significant advances are being made in cancer biology, which includes deciphering some specifics on how TLRs play a key role in anti-cancer immunity and cancer rejection. As exogenous and endogenous TLR ligands are crucial for anti-cancer immunity, there is an urgent need to develop novel TLR-stimulating therapies. Meaningful research is needed to assuredly elucidate the roles of TLRs in modulating cancer immunotherapy and their clinical outcomes. In this regard, the majority of the currently investigated TLR agonists as anti-cancer targets are non-protein microbial components, such as lipopolysaccharides, oligonucleotides, and lipopeptides. However, a growing number of studies reveal that TLR signaling and subsequent immune responses can be activated by numerous microbial proteins and antigens.

While we still need to understand the TLR–TLR ligand pairs to produce desired oncological outcomes, there is growing evidence to suggest that TLR-modulating therapies will prove to be a safe treatment for some types of cancers. Since cancers are heterogeneous in nature, multi-treatment regimens may be useful, in some cases, in addition to known immunotherapies. TLRs are known to play a complex functional role in tumor biology and, at times, act as a double-edged sword in immunotherapy.

Recently it has become evident that TLRs do not differ from other immune receptors in their compliment to launch both host defenses and cell death. Our understanding of the signaling cascades starting from TLR activation down to cell activation has significantly progressed during the last decade. However, the molecular pathways leading to TLR-induced apoptosis are yet to be addressed.

Extensive research is required to determine the combinatorial use of TLR ligands that will prove to be smarter therapies with less toxicity and improved potency. Further studies on the roles of TLRs and their functions as anti-cancer immunity and cancer rejection will nobly advance the development of therapeutic interventions and will benefit patients undergoing immunotherapy.

## Figures and Tables

**Figure 1 cancers-14-02923-f001:**
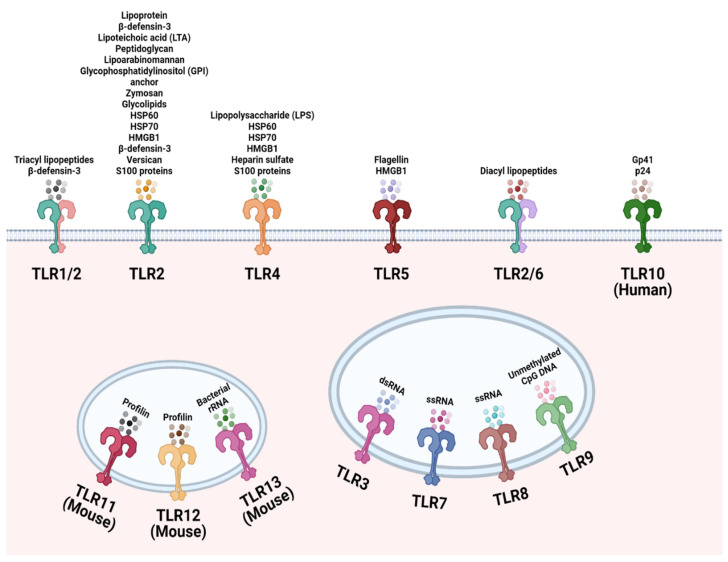
Cellular localization of the Toll-like receptor (TLR) family. TLR1, TLR2, TLR4, TLR5, TLR6, and TLR10 are localized to the cell surface to recognize common microbial structural components and endogenous ligands. TLR3, TLR7, TLR8, TLR9, TLR12 and TLR13 are located on endosomes to sense microbial nucleic acids that have entered the cell. TLR11 and TLR12 are localized to endosomes to recognize *Toxoplasma gondii* derived profilin. Created with Biorender.com.

**Figure 2 cancers-14-02923-f002:**
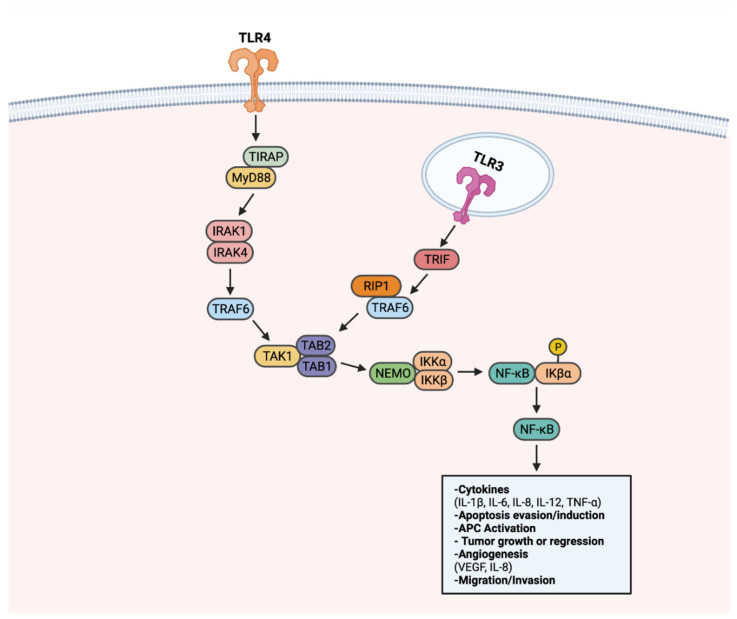
TLR-NF-κB signaling in progression or regression of cancer. Upon recognition of a ligand, TLRs transduce signals through a MyD88-dependent signaling pathway, or alternatively a MyD88-independent TRIF-activating pathway. Upon activation of the MyD88-dependent pathway, adapter molecule TIRAP transduces signals to MyD88. MyD88 activates IRAK1 and IRAK4. Activated IRAK1 and IRAK4 lead to activation of TRAF6. TRAF6 activates the TAK1 complex. The TAK1 complex activates the IKK complex comprised of NEMO, IKKα, and IKKβ. IKK complex activation leads to phosphorylation of IKβα, a protein responsible for sequestering NF-κB to the cytoplasm. Once activated, NF-κB translocates to the nucleus to activate genes that can promote or inhibit tumorigenesis. NF-κB also can be activated through TRIF, notably through TLR3. TRIF activation results in RIP1 and TRAF6 activation. Through RIP1 and TRAF6, the TAK1 complex is activated. Following TAK1 complex activation, subsequent steps in NF-κB signaling are shared between the two pathways. Created with Biorender.com.

**Figure 3 cancers-14-02923-f003:**
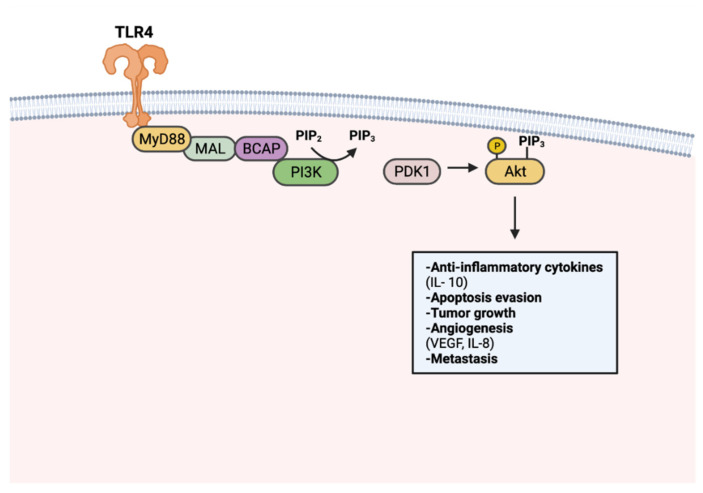
TLR-mediated PI3k-Akt signaling in tumorigenesis. PI3k is activated upon TLR stimulation through MyD88, MAL, and BCAP. Activated PI3k converts phosphatidylinositol 4,5-bisphosphate (PIP_2_) to phosphatidylinositol (3,4,5)-trisphosphate (PIP_3_). Akt activation occurs through PIP_3_-facilitated recruitment to the plasma membrane and phosphorylation by PDK1. Activated Akt promotes tumor progression through anti-inflammatory cytokine production, apoptosis resistance, tumor growth, angiogenesis, and metastasis. Created with Biorender.com.

**Figure 4 cancers-14-02923-f004:**
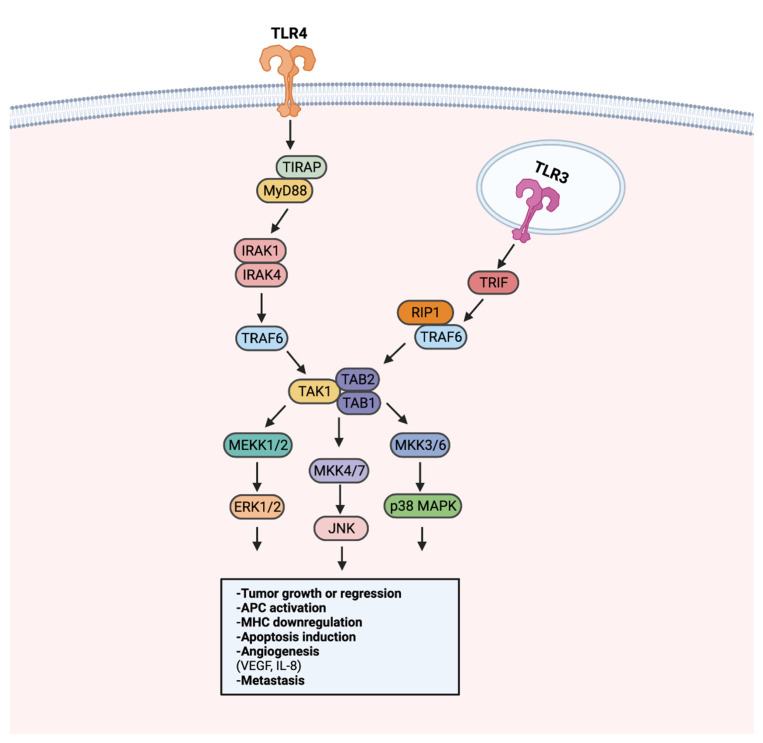
TLR-MAPK/ERK signaling in the progression or regression of cancer. MAPK/ERK pathway activation can occur through MyD88-dependent or MyD88-independent TRIF signaling. The MyD88-dependent and -independent signaling converges in the activation of the TAK1 complex. Following the activation of the TAK1 complex, MEKK1/2, MKK4/7, and MKK3/6 are activated. MEKK1/2, MKK4/7, and MKK3/6 activation leads to the activation of ERK1/2, JNK, and p38 MAPK, respectively. Signaling from ERK1/2, JNK, and p38 MAPK can promote tumor growth or inhibition, APC modulation, angiogenesis, and metastasis. Created with Biorender.com.

**Figure 5 cancers-14-02923-f005:**
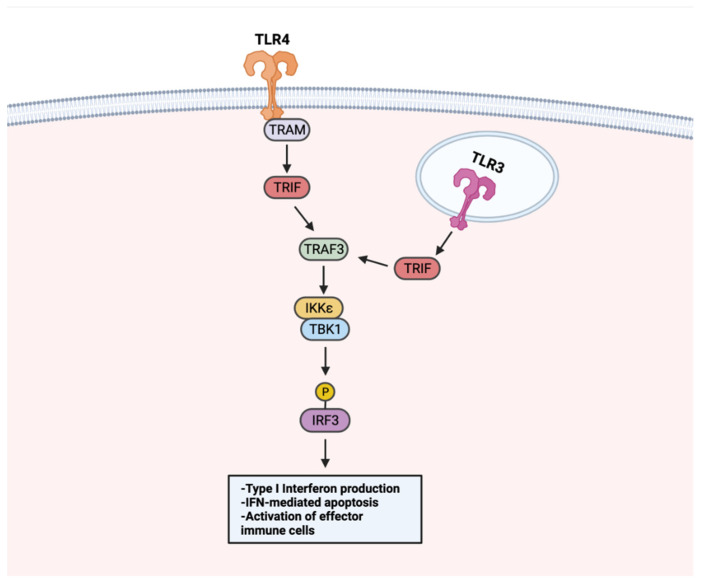
TLR–IFN signaling in anti-tumor responses. TLRs use a MyD88-independent pathway for IFN signaling. Upon TLR stimulation, membrane-localized TLRs (i.e., TLR4) activate TRIF through adaptor protein TRAM. TRIF activates TRAF3, which activates the TBK1–IKKε complex. The TBK–IKKε complex phosphorylates IRF3. Endosome-localized TLR3 activates TRIF directly. Following TRIF activation, TRAF3 is activated and shares downstream IFN signaling steps with membrane-localized TLRs. TLR–IFN signaling induces type I IFN production, apoptosis, and activation of immune cells. Created with Biorender.com.

**Table 1 cancers-14-02923-t001:** TLR Expression and Tumor Outcomes.

Cancer Type		TLR-Cell Line Characterization	Pre-Clinical Findings	Tumor Profile and Patient Outcomes	References
**Breast Cancer**	**TLR2**	MDA-MB-231, SUM-149, SUM-159	-	Expression observed in primary tumors and metastatic tissue; high expression associated with shorter overall survival	[18,19,20]
	**TLR3**	MDA-MB-231, MDA-MB-468, SUM-149, SUM-159	Poly(I:C) stimulation reduces breast cancer cell proliferation and induces apoptosis	Upregulated in recurring tumors; associated with lower relapse-free survival	[19,21,22]
	**TLR4**	MDA-MB-231, SUM-149, SUM-159	LPS stimulation induces IL-6 and IL-9 production; activation promotes chemoresistance and apoptosis evasion; downregulation enhances paclitaxel sensitivity; upregulation promotes paclitaxel resistance	Expression observed in primary tumors and metastatic tissue; upregulation associated with tumor recurrence and poor survival in TP53 mutant tumors	[18,19,21,23,24]
	**TLR5**	4T1	Downregulation upregulates VEGFR and cell proliferation; upregulation and downregulation of receptor increases lung metastases; flagellin treatment reduces tumor growth	Highly expressed in metastatic cancer	[18,25,26,27]
	**TLR7/8**	-	-	Low expression observed in metastases; imiquimod promotes immune cell infiltration in skin-residing metastases	[18,28,29]
	**TLR9**	MCF-7, T47D, CAMA, MDA-MB-231, MDA-MB-468, SUM-149, SUM-159	Receptor knockdown promotes MDA-MB-231 tumor growth	Expression observed on tumors; low expression in metastases; downregulation associated with poor disease-specific survival	[18,19,21,30]
**Lung Cancer**	**TLR2**	-	Treatment with lipoprotein reduces Lewis lung carcinoma tumor growth	High tumoral TLR2 expression is positively correlated with prolonged overall survival and progression-free survival	
	**TLR3**	Calu-3; H460	Lewis lung carcinoma tumors in TLR3-deficient mice had fewer metastases compared to TLR3 competent mice; stimulation with Poly(I:C) induces apoptosis	TLR3 positive tumors have greater overall survival and slower disease progression in early-stage NSCLC	[26,31,32,33]
	**TLR4**	A549; H1299	Stimulation with LPS induces production of TGF-β, VEGF, and IL-8	High expression associated with decreased overall survival; expression correlated with tumoral PD-L1 expression	[28,34,35]
	**TLR5**	SPC-A1; A549; H1975; H1299	Stimulation with flagellin induces IL-6 and CXCL5 production	High expression associated with improved disease-free survival	[36]
	**TLR7/8**	A549, H1355, SK-MES; LL/2	Stimulation promotes survival and chemotherapy resistance, CL264 treatment enhances Lewis lung carcinoma tumor growth; resiquimod formulation improves overall survival and reduced 344SQ tumor progression	High expression associated with poor overall survival in stage I-III NSCLC patients	[27,37,38]
	**TLR9**	A549, NCI-H727	Expressed on human NSCLC cell line A549; synthetic oligonucleotide treatment reduces tumor growth in H520, H358, A549, and H1299 xenografts	Higher expression in tumors compared to non-cancerous tissue	[39,40]
**Melanoma**	**TLR2**	ME5, ME9, ME16, ME17, ME19	Stimulation promotes cell migration; treatment with Zymosan-A and bacteria reduces B16-F10 tumor growth	Expression observed on tumors	[41,42,43]
	**TLR3**	ME2, ME9, ME16, ME17, ME19, M288, M301, M305, M299, M342	Stimulation promotes cell migration	Expression observed on tumors	[42,43,44]
	**TLR4**	ME2, ME9, ME16, ME17, ME19	Stimulation promotes cell migration	Highly expressed on primary and metastatic tumors; expression associated with shortened relapse-free survival	[42,43,44,45]
	**TLR7/8**	M288, M301, M305, M284, M379, M299, M342, M383, M350, M383, M387	Imiquimod stimulation inhibits tumoral angiogenesis in a melanoma-bearing humanized mouse model	Upregulated expression in stage III melanoma patients; high expression associated with longer overall survival time; expression correlated with CD8+ T-cell infiltration; treatment with imiquimod inhibits metastasis	[43,44,46,47]
	**TLR9**	M288, M301, M305, M350, M387	Treatment with L-nucleotide-protected TLR agonists reduce B16-F10 tumor growth	Expression observed on tumors	[43,44,48]
**Colorectal Cancer**	**TLR1/2**	-	-	Upregulated in cancerous tissue; high expression associated with improved disease-specific survival	[49,50,51]
	**TLR3**	HCT116, HT29, SW620	Poly(I:C) stimulation induces CCL2, CCL5, and IL-8 production; CXCL8 production, and invasiveness in CRC cell lines	Low expression associated with lymph node metastasis and tumor recurrence	[52,53]
	**TLR4**	-	Upregulated in chemically induced CRC in Tir8 −/− mice	Expression upregulated in cancerous tissue; high expression associated with poor disease-free survival	[54]
	**TLR5**	DLD-1	Knockdown promotes DLD-1 tumor growth and inhibits immune cell infiltration	Low expression associated with advanced cancer stage; high expression associated with improved disease-specific survival	[50,55]
	**TLR7/8**	-	R848 treatment of CT26 tumors reverses chemoresistance to oxaliplatin	Upregulation observed in tumors, associated with lower cancer stage; high expression associated with improved disease-specific survival	[50,51,56]
	**TLR9**	-	Stimulation reduces CT26 tumor growth, increases CD8+ T-cell infiltration in the tumor	High expression correlated with invasiveness, metastasis, and advanced-stage CRC	[48,57]
**Pancreatic Cancer**	**TLR2**	HPAC, MIA PaCa-2, PANC-1, BXPC-3, PaCaDD135	Stimulation promotes cell proliferation, VEGF expression, and colony formation	Highly expressed in all stages of PDAC; upregulation correlated with poor patient survival	[58,59]
	**TLR3**	PANC-1, BXPC-3	Activation promotes cell proliferation	-	[60,61]
	**TLR4**	MIA PaCa-2, SW1990	LPS stimulation mediates tumorigenesis in p48Cre;Kras^G12D^ mice, and promotes cell proliferation and VEGF expression	Upregulated in cancerous tissue	[59,62]
	**TLR7/8**	PANC-2	Stimulation promotes cell proliferation, chemoresistance, and tumorigenesis in p48Cre;Kras^G12D^ mice; inhibition prevents tumor progression	Expression upregulated in early and advanced stages of PDAC	[63,64]
	**TLR9**	PANC-1, SW1990, PaCaDD185, PAN02	Stimulation promotes cell proliferation, VEGF expression, and tumorigenesis in p48^Cre^;LsL-Kras^G12D^ mice; inhibition improves survival and prevents tumor progression	Upregulated in cancerous tissue	[59,65]
**Ovarian Cancer**	**TLR2**	SKOV3, CAOV3	Expression upregulated upon tumor injury in xenografted mice; activation promotes tumoral repair and persistence	Upregulated in cancerous tissue	[66,67]
	**TLR3**	ES2, OVCAR3, SKOV3, CAOV3	Stimulation induces CCL2 and IL-6 production	Upregulated in cancerous tissue	[67]
	**TLR4**	R182, CP70, A2780, R179, OVCAR3, SKOV3, AD-10, ES2	Stimulation promotes cancer cell viability and cell proliferation and induces CCL2, IL-6, and CXCL1 production; knockdown enhances sensitivity to paclitaxel	High expression in cancerous tissue; high expression associated with improved survival	[67,68,69,70,71]
	**TLR5**	OVCAR3	TLR5-deficiency reduces tumor growth; stimulation promotes invasion	Polymorphism diminishing TLR5 signaling improves long-term survival	[72,73]
	**TLR7/8**	CaOV3, OVCAR3, OV90, SKOV3	Stimulation promotes invasion	-	[72,74]
	**TLR9**	-	-	Increased expression associated with rising tumor grade	[75]
**Prostate Cancer**	**TLR2/6**	PC3	Stimulation promotes cell proliferation and invasiveness and induces IL-6 and IL-8 production	-	[76,77]
	**TLR3**	LNCaP, DU145, PC3	Stimulation inhibits cell proliferation and promotes apoptosis and induces IL-8, CCL3, CCL5, and CXCL10 production	Upregulated in cancerous tissue; high expression associated with poor patient survival	[23,76,77,78]
	**TLR4**	PC3, DU145	Stimulation promotes cell proliferation and induces IL-6 and IL-8 production; knockdown diminishes tumorigenesis, reduces cell invasiveness and proliferation, and induces apoptosis	Upregulated in cancerous tissue	[23,76,77,79]
	**TLR5**	DU145, PC3, LNCaP	Stimulation induces IL-8 and CCL5 production	-	[76]
	**TLR9**	-	-	Upregulated in cancerous tissue; high expression associated with poor patient survival	[79,80]

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
