# Peer review of "Microbial-Derived Toll-like Receptor Agonism in Cancer Treatment and Progression"

_cancers, 2022, doi:10.3390/cancers14122923_

Round 1
Reviewer 1 Report
This is a timely and well-organized review article. I recommend publishing this article with the following modifications and/or discussion:
It will be interesting to have a discussion on small-molecule TLR agonists. Is there any specific reason to omit them?
Author Response
Comment 1: This is a timely and well-organized review article. I recommend publishing this article with the following modifications and/or discussion:
We are grateful to the reviewer for their kind words and for taking their time to review our article.
Comment 2: It will be interesting to have a discussion on small-molecule TLR agonists. Is there any specific reason to omit them?
Thank you for this insight on including small molecules TLR agonists.
While small-molecule TLR agonists indeed have been utilized in anticancer therapies and have been well-reviewed in past, many of those agonists are synthetically derived. We believe that highlighting these agonists would stray from the focus of this review on microbial-based TLR agonists.
Thank you,
Kajal
Reviewer 2 Report
The review article by Giurini et al. discussed the TLRs expression in cancer types and its clinical relevance. Authors have included the studies that highlighted the pro and anti-tumor role of TLRs signaling in various types of cancers. Moreover, the authors have nicely summarized most of the studies in Table 1. I have some suggestions for the authors given below.
Major -
1-Please also include a Table for immune cells including expression pattern of TLRs, agonists used for treatment, cancer types, mechanism of action, and outcome. It will help to understand the role of TLRs signaling in tumor and immune cells.
2- Please add some studies related to TLR- mediated metabolic reprogramming of tumors and immune cells with its implications on cancer therapy.
Minor-
1- In Figures 2 and 3, please show the MyD88- independent TLRs signaling as discussed in figure legends. If possible, please also include the other TLR’s name in the figure as they shared the common pathways
Author Response
The review article by Giurini et al. discussed the TLRs expression in cancer types and its clinical relevance. Authors have included the studies that highlighted the pro and anti-tumor role of TLRs signaling in various types of cancers. Moreover, the authors have nicely summarized most of the studies in Table 1. I have some suggestions for the authors given below.
Ans: We are grateful for the reviewer's kind words, we have done in-depth studies on TLR pathways and agonists to write this article.
Major Comment 1: Please also include a Table for immune cells including expression pattern of TLRs, agonists used for treatment, cancer types, mechanism of action, and outcome. It will help to understand the role of TLRs signaling in tumor and immune cells.
Ans: We opted to not include a table focused solely on TLR expression on immune cells, as this has been greatly highlighted in a number of other review articles. However, we have covered the TLR agonists used for treatment and cancer types inTable 1, which encompasses multi-TLR expression on seven different cancer types, as well as the effects of activation of a given TLR on a given type of cancer the in vitro, in vivo, and in patient outcome.
Major Comment 2: Please add some studies related to TLR- mediated metabolic reprogramming of tumors and immune cells with its implications on cancer therapy.
Ans: This is a great suggestion and something we are looking into for our future studies. However, TLR-mediated metabolic reprogramming in the context of cancer warrants further examination, this falls outside the scope of our current review article.
Minor Comment 1: In Figures 2 and 3, please show the MyD88- independent TLRs signaling as discussed in figure legends. If possible, please also include the other TLR’s name in the figure as they shared the common pathways.
Ans: We have included both MyD88-dependent and MyD88-independent TLR signaling, as demonstrated by membrane-localized TLR4 and endosomal localized TLR3, respectively, in figure 2.
MyD88-independent signaling was not included in Figure 3, as TLR-mediated MyD88-independent PI3K-Akt signaling has not been properly studied, we did not wish to include studies that are not properly cited. In addition, TLR-mediated signaling has been most thoroughly characterized through TLR4, as it is also the most well-studied Toll-like receptor.
Thank you,
Kajal
Reviewer 3 Report
The review is in a good shape. It presents good updated information about the area of study and clinical assays.
I suggest minor changes that may need to be addressed. Citation format, some present superscript (highlighted in yellow).
Biorender figures need to have the same font letter as the rest of the paper.

Author Response
The review is in a good shape. It presents good updated information about the area of study and clinical assays.
I suggest minor changes that may need to be addressed. Citation format, some present superscript (highlighted in yellow).
Biorender figures need to have the same font letter as the rest of the paper.
Ans: We are grateful for the reviewer's comments and suggestions. We have made appropriate changes to the citation format.
Thank you,
Kajal
Round 2
Reviewer 2 Report
Dear authors,
Thanks for your clarification, you only want to focus on tumor cells.